# Conjugation Invariant Learning with Neural Networks

## Abstract

Machine learning under the constraint of symmetries, given by group invariances or equivariances, has emerged as a topic of active interest in recent years. Natural settings for such applications include the multi-reference alignment and cryo electron microscopy, multi-object tracking, spherical images, and so on. A fundamental paradigm among such symmetries is the action of a group by symmetries, which often pertains to change of basis or relabelling of objects in pure and applied mathematics. Thus, a naturally significant class of functions consists of those that are intrinsic to the problem, in the sense of being independent of such base change or relabelling; in other words invariant under the conjugation action by a group. In this work, we investigate such functions, known as class functions, leveraging tools from group representation theory. A fundamental ingredient in our approach are given by the so-called irreducible characters of the group, which are canonical tracial class functions related to its irreducible representations. Such functions form an orthogonal basis for the class functions, extending ideas from Fourier analysis to this domain, and accord a very explicit structure. Exploiting a tensorial structure on representations, which translates into a multiplicative algebra structure for irreducible characters, we propose to efficiently approximate class functions using polynomials in a small number of such characters. Thus, our approach provides a global, non-linear coordinate system to describe functions on the group that is intrinsic in nature, in the sense that it is independent of local charts, and can be easily computed in concrete models. We demonstrate that such non-linear approximation using a small dictionary can be effectively implemented using a deep neural network paradigm. This allows us to learn a class function efficiently from a dataset of its outputs.

## 1 Introduction

**Learning under group actions.** Learning under group actions has emerged as a topic of key interest in recent years. The natural motivation for this paradigm comes from the many applications in which there is a canonical action of a group of symmetries, often arising as isometries or invariances of the model under consideration (Jebara & Kondor, 2008; Wein, 2018).

Instances of learning models that admit natural group actions are manifold. Examples range from natural image data that exhibit symmetries accorded by rotations and translations (Kondor, 2007; Kakarala, 2012); data arising in tomography (Vasco, 2007; Moroder et al., 2012); network based data (Giridhar & Kumar, 2006; Herzig et al., 2018); synchronization problems in robotics (Rosen et al., 2020); computer vision (Agrawal et al., 2006); problems like multi object tracking where many natural questions are invariant under the rearrangements of object labels (Kondor et al., 2007); atomic physics (Seko et al., 2019); learning permutations (Huang et al., 2009); Gaussian mixture models under group actions (Brunel, 2019), among many others (Bandeira et al., 2015b). Significant recent trends include neural networks that are equivariant under group actions on the data (Kondor & Trivedi, 2018; Worrall et al., 2016; Weiler et al., 2018a; Thiede et al., 2021) and related convolutional neural networks (CNNs) (Kondor et al., 2018; Cohen et al., 2018b; Niepert et al., 2016; Defferrard et al., 2016; Cohen & Welling, 2016; Cohen et al., 2018a; Esteves et al., 2018; Weiler et al., 2018b); multi-reference alignment (MRA) (Bandeira et al., 2014; Perry et al., 2019; Bendory et al., 2017; Sorzano et al., 2010; Abbe et al., 2018; Fan et al., 2020) and cryo electron microscopy (cryo EM)

under the action of the relevant Euclidean isometries (Singer et al., 2018; Hadani & Singer, 2011; Bendory et al., 2020; Bandeira et al., 2015a); and so forth.

**Invariant functionals.** A significant class of observables (i.e., functionals of the data) that attract natural interest in this setting consist of those that are invariant with respect to the group action under consideration; e.g., mutual distances under the action of a group of isometries, or the relative order of two objects under the permutation of their labels. In this work, we focus on a fundamental category of such observables, called *class functions*, that are invariant under the conjugation action of a group.

**The conjugation action.** Group action by conjugation is one of the foundational ways in which a group can act, and usually corresponds to a change of basis, or relabelling of objects or indices. A simple example is accorded by data in the form of matrices, where conjugation by an invertible matrix translates into a change of Euclidean basis, leading to the presentation of the information in a different co-ordinate system. In this setting, any function of the spectrum of the data matrix remains invariant under the change of basis (i.e., the conjugation action), and is clearly a significant feature to study. Yet another ready example is accorded by the permutation of the node labels in a network, which leaves invariant important network properties, such as diameter, subgraph counts and other aspects of network geometry, as well as the spectral properties of the network.

**Symmetries, group representations and Fourier analysis.** Fourier analysis is a central tool in fundamental machine learning tasks, entailing a vast collection of concepts including bandlimited functions and low pass filters, productively focussing on the low frequency part of the Fourier spectrum of a signal as the most significant component. At a conceptual level, much of this originates from the actions of the translational group of symmetries acting canonically on signals defined on Euclidean spaces; in particular, the structural simplicity is largely a consequence of the fact that these groups are *abelian* or *commutative*.

However, natural learning scenarios where the canonical symmetries form a *non-abelian* group (i.e., *non-commutative*) are numerous; even on Euclidean spaces, the translational symmetries when augmented with rotations constitute a ready example. In such settings, *group representation* theory (Simon, 1996) accords an extension of many of the Fourier analytic concepts; for instance the expansion of a function in terms of *matrix-valued Fourier coefficients* indexed by the so-called *irreducible representations* of the group. For a more detailed discussion on the fundamental synergies between Fourier analysis and group representations, we refer the reader to Sec. E in the appendix.

**A non-linear conjugation invariant learning paradigm with neural networks.** In the present work, we are focussed on learning conjugation invariant functions (i.e., class functions), which happen to accord a much simpler version of the non-commutative Fourier analysis discussed above. This is realised via an orthogonal linear basis of the so-called *irreducible characters*, which are essentially traces of certain minimal matrix representations of the group, and are well-understood in many concrete applications (c.f. Sec. E in appendix for details). It turns out that the space of irreducible characters exhibit a much richer structure in the form an *algebra*, which entails that it is closed under polynomial combinations. As an upshot, using polynomials of a few irreducible characters of relatively low order, one is able to express much more complicated characters of substantially higher order; leading to an approximation paradigm for general class functions via non-linear, polynomial combinations of a small alphabet of irreducible characters.

In implementing our programme, a fundamental role is played by neural networks, which allow us to effectively implement this non-linear approximation scheme. Our approach, in essence, leads to a global co-ordinate system for learning conjugation invariant functions on groups. Viewed from the perspective of function space approximation, our paradigm may be seen as a kind of non-linear dimension reduction tool for class functions.

**Groups acting on homogeneous spaces.** In some applications, the group of symmetries $G$ acts on a set $X$ of objects (e.g., rotations acting on a sphere) ; such a set (under a mild *transitivity* assumption) is referred to as a homogeneous space of the group. Our setup may be extended to cover such settings, using a canonical 'lift' of any function $f : X \mapsto \mathbb{R}$ to a function $\tilde{f} : G \mapsto \mathbb{R}$ that satisfies $\tilde{f} = f \circ \pi$, for a standard quotient map $\pi : G \mapsto X$ (Dummit & Foote, 2004). Stipulating that $\tilde{f}$ is conjugation-invariant is equivalent to $f$ being $[G, G]$-invariant on $X$, where $[G, G]$ is the so-called *commutator* subgroup of $G$ (c.f. Lemma C.1 in appendix). This covers the setting of functions $f$ that are constant on the orbits of $[G, G]$ acting on $X$. Such functions include binary classifiers on the permutation group $S_n$, classification problems on the dihedral group $D_{2n}$, and so on. For a detailed

discussion, we refer the reader to Sec. C in the appendix. Further applications of our paradigm to groups acting on homogeneous spaces can be obtained via a general averaging paradigm, e.g. in the context of image alignment problems (c.f. Proposition 2.1).

**Related literature.** Class functions have appeared in various forms as useful methodological tools in the machine learning literature. These include the study of bi-invariant kernels in the context of RKHS on groups and homogeneous spaces (Jebara & Kondor, 2008); applications to problems of multi-object tracking Kondor et al. (2007); learning permutations (Jebara & Kondor, 2008); diffusion kernels on graphs Kondor & Lafferty (2002); Saloff-Coste (2001), and so on. Yarotsky (2021) examines an invariant theory based approach to learning under general symmetries via two layer neural nets with rather complex polynomial inputs; in contrast our approach inputs the ubiquitous group characters and approximates polynomials of high degree, via a few-layer network architecture that exploits the compositional aspect of neural nets in a non-trivial way. To the best of our knowledge, our approach to efficient learning of conjugation invariant functions through the lens of group character theory appears to be rather novel, and prior literature in a similar vein appears to be highly limited.

## 2 CONJUGATION INVARIANT FUNCTIONS AND THEIR APPLICATIONS

Our interest in this paper would be focussed on the so-called *class functions*, so we begin with a concrete definition.

**Definition.** Let $G$ be a group. Two elements $x, y \in G$ are said to be *conjugate* if $\exists t \in G$ such that $txt^{-1} = y$. A function $f : G \mapsto S$ for any set $S$ is called a *class function* if $f(txt^{-1}) = f(x) \quad \forall t \in G$.

Conjugacy is an equivalence relation, partitioning $G$ into equivalence classes which are referred to as conjugacy classes. It follows from the definition that conjugation-invariant functions are constant on these conjugacy classes, hence the name class functions.

**Class functions on SO(3) and $S_n$.** Elements of the 3-D rotation group SO(3) are precisely $3 \times 3$ orthogonal matrices with determinant equal to $1$. Two elements of SO(3) are conjugate if and only if they are similar matrices. This means that class functions on SO(3) only depend on the matrix spectrum of the elements of SO(3). In fact, every $R \in$ SO(3) is similar (via conjugation) to a block diagonal matrix of the form $\begin{pmatrix} \cos\theta & -\sin\theta & 0 \\ \sin\theta & \cos\theta & 0 \\ 0 & 0 & 1 \end{pmatrix}$. In geometric terms, the conjugacy classes of SO(3) are the sets of rotations by the same angle $\theta$ about *different axes*. The trace of these matrices is an example of a class function; $\mathrm{tr}(R) = 1 + 2\cos\theta$.

In the case of the symmetric group $S_n$, two permutations lie in the same conjugacy class if and only if they have the same cycle shape. Since cycle shapes are conveniently labelled using partitions, the conjugacy classes of $S_n$ are most commonly labelled using partitions of $n$.

Class functions arise in many natural settings in machine learning. We provide below several such scenarios where class functions are of key importance.

**A general averaging paradigm.** We first provide a general setting where class functions arise naturally through a process of averaging over the group; conjugation invariant functionals often emerge in practical problems through such a procedure. Let $G$ be a group acting on a space $X$; that is for every $g \in G$, we may associate a map $T_g : X \to X$ such that $T_e = \mathrm{id}_X$ is the identity map on the space $X$ (where $e$ is the identity element in the group $G$), and $T_{gh} = T_g \circ T_h \quad \forall g, h \in G$. Let $\mu$ be the left Haar measure on the group $G$, i.e., $\mathrm{d}\mu(g \cdot h) = \mathrm{d}\mu(h)$ for all $g, h \in G$. Then we may state:

**Proposition 2.1.** *Let $f : X \times X \mapsto \mathbb{R}$ be such that $f(g(x), g(y)) = f(x, y) \, \forall x, y \in X, g \in G$. For any pair $x, y \in X$ and $T \in G$, define $\tilde{f}_{x,y}(T) := \int_G f(Tg(x), g(y)) \mathrm{d}\mu(g)$. Then $\tilde{f}_{x,y}$ is a class function.*

Functions $f$ as in Proposition 2.1 that are invariant under the diagonal action of $G$ naturally occur in a wide array of settings; prominent among them being in the form of distances (or angles) on a Euclidean space $X$ under the action of a group of isometries $G$. For a concrete use-case, we refer to the image alignment application below.

*Proof of Proposition 2.1.* Given any $h, T \in G$, we have

$$\tilde{f}_{x,y}(hTh^{-1}) = \int_G f(hTh^{-1}g(x), g(y))d\mu(g) = \int_G f(Th^{-1}g(x), h^{-1}g(y))d\mu(g)$$

$$= \int_G f(Tg'(x), g'(y))d\mu(g') = \tilde{f}_{x,y}(T).$$

The third equality is due to left-invariance of $\mu$ and setting $g' = h^{-1}g$. $\qquad\square$

**Image Alignment.** Image alignment problems have received considerable attention in recent years, and furnish a natural example of class functions. To set the stage, we consider spherical images, which are well-motivated by projections of planar images onto the northern sphere (Kondor et al., 2018; Jebara & Kondor, 2008), images in the form of photos taken using a $360°$ camera, and so on. In the setting of Proposition 2.1, we set $G = \text{SO}(3)$, $X = S^2$ and $f(x, y)$ to be the Euclidean distance between $x$ and $y$, which would naturally be rotation invariant. Then $\tilde{f}_{x,y}(T)$ is a measure of how well an image centered at $x$ is aligned with an image centered at $y$ after applying the rotation operator $T$. The quantity $\mathcal{M}_{x,y} = \min_T \tilde{f}_{x,y}(T)$ captures the maximal alignment between the images, and can be used e.g. for classification tasks (e.g., if $x$ is a prototype and $y$ is a new image, then we classify $y$ to the category of $x$ if $\mathcal{M}_{x,y}$ smaller than a threshold). Proposition 2.1 demonstrates that $\tilde{f}_{x,y}$ is a class function on $SO(3)$.

**Quadratic Assignment Problem.** Let $P_n$ denote the group of $n \times n$ permutation matrices; that is matrices having exactly one entry equal to 1 in each row and column, and the remaining entries being 0. The *quadratic assignment problem (QAP)* is a fundamental problem in combinatorial optimization Koopmans & Beckmann (1957), where the goal is to find $\arg\min_{X \in P_n} \text{tr}(AXBX^T)$; we will consider this problem in the setting where $A \in P_n$ and $B$ is a general $n \times n$ matrix.

**Proposition 2.2.** *Let $f : P_n \mapsto \mathbb{R}$ be given by $f(A) := \min_{X \in P_n} \text{tr}(AXBX^T)$, and $B$ be an $n \times n$ matrix. Then $f$ is a class function.*

*Proof.* We verify that $f$ is a class function. Fix any $Q, A \in P_n$. Note that $Q^T = Q^{-1}$.

$$f(QAQ^{-1}) = \min_{X \in P_n} \text{tr}(QAQ^T XBX^T) = \min_{X \in P_n} \text{tr}(Q^T(QAQ^T XBX^T)Q)$$

$$= \min_{X \in P_n} \text{tr}(AQ^T XBX^T Q) = \min_{X \in P_n} \text{tr}(A(Q^T X)B(Q^T X)^T)$$

$$= f(A).$$

$\qquad\square$

This gives us a *stopping algorithm* (or target) for QAP in the following sense. Once we learn the function $f$, we could obtain or approximate the objective minimum $f(A)$ for the QAP. We can then use some optimization algorithm (such as gradient descent) to update the matrix $X$ in order to decrease the objective $\text{tr}(AXBX^T)$, and terminate the optimization process once $\text{tr}(AXBX^T) - f(A) < \epsilon$ for some acceptable error tolerance threshold $\epsilon > 0$.

**Testing on comparative rankings.** Denote $\{1, 2, \ldots, n\}$ by $[n]$. Let $S_n$ be the symmetric group on $[n]$. Suppose that we have two real-valued functions $f_1, f_2 : [n] \to \mathbb{R}$. Then, $f_1$ and $f_2$ would induce orderings $\sigma_1$ and $\sigma_2$ respectively of $[n]$. For instance, we could have $n$ individuals and $f_1(k), f_2(k)$ are resp. the height and weight of individual $k$, leading to their ranking by heights or weights (denoted resp. by $\sigma_1, \sigma_2$). A natural question is the comparative analysis of such rankings, e.g. testing whether they are approximately similar. In such considerations, of natural interest are statistics like the number of individuals having the same rank, given by $\mathcal{L}_n := \frac{1}{n} \sum_{k \in [n]} \mathbf{1}_{[\sigma_1(k) = \sigma_2(k)]} = \frac{1}{n} \sum_{k \in [n]} \mathbf{1}_{[\sigma_2^{-1}\sigma_1(k) = k]}$, where $\mathbf{1}$ is the indicator function.

Let $f : S_n \to \mathbb{R}$ be the function $f(\sigma) = \frac{1}{n} \sum_{k \in [n]} \mathbf{1}_{[\sigma(k) = k]}$ for all $\sigma \in S_n$; notice that $\mathcal{L}_n = f(\sigma_2^{-1}\sigma_1)$. For any $\pi, \sigma \in S_n$, we may compute $f(\pi\sigma\pi^{-1}) = \frac{1}{n} \sum_{k \in [n]} \mathbf{1}_{[\pi\sigma\pi^{-1}(k) = k]} = \frac{1}{n} \sum_{k \in [n]} \mathbf{1}_{[\sigma\pi^{-1}(k) = \pi^{-1}(k)]} = f(\sigma)$. This demonstrates that the important statistic $\mathcal{L}_n = f$ is a class function.

**Multi-object tracking.** In the multi-object tracking problem, there are $k$ objects with labels $\{1, \ldots, k\}$, and we observe their arrangement which evolves with time; e.g. the evolution of the formation of jets flying in an air show. The data is succinctly described in terms of a permutation $\sigma(t)$ mapping each label to its position in the arrangement at time $t$. Most of the fundamental questions about this model are actually independent of the labels; e.g., whether the starting and the final arrangements are the same, or the first time $t = t_0$ at which $\sigma(t)$ undergoes a change. etc. Any change in object labels amounts to a conjugation $\sigma(t) \mapsto \pi\sigma(t)\pi^{-1}$ for some permutation $\pi$, leading to the fact that the answer to the above natural questions pertaining to relative order are all class functions.

## 3 TENSOR PRODUCTS AND THE ALGEBRA OF CHARACTERS

The theoretical foundations of our approach are underlaid by the theory of tensor products of group representations, which manifests itself in the form of a polynomial algebra structure on group characters. Below, we provide a rudimentary outline of this theory, focussing on aspects that are fundamental to our approach. For an account of group representation theory essential for our approach, we refer the reader to Sec. E in the appendix.

The *character* $\chi_\rho$ of a matrix representation $\rho$ of a group $G$ is the function $\chi : G \to \mathbb{C}$ defined by $\chi(x) = \text{tr}(\rho(x))$. The characters inherit the inner product from the $L^2$ structure on the group $G$. We call $d_\rho$ the *order* of $\rho$ or $\chi_\rho$. If $\rho$ is an *irreducible* representation, we say that $\chi_\rho$ is an *irreducible character*. Since $\text{tr}(T^{-1}AT) = \text{tr}(A)$, it is evident that $\chi_\rho$ is a class function. By the celebrated Peter-Weyl theorem (Simon, 1996), the set of irreducible characters form an orthonornal basis for the space of square integrable class functions.

Let $\rho_1$ and $\rho_2$ be two representations of $G$ with corresponding characters $\chi_1$ and $\chi_2$ respectively. The tensor product representation is defined as $\rho_1 \otimes \rho_2(g) := \rho_1(g) \otimes \rho_2(g)$ for all $g \in G$ (to be understood in the form of tensor products of matrices). In general, if $\rho_1 \otimes \rho_2$ is not an irreducible representation of $G$ even if $\rho_1$ and $\rho_2$ are irreducible. However, the theorem of complete reducibility (Simon, 1996) tells us that it must decompose uniquely as a direct sum of irreducible representations of $G$.

$$\rho_1(g) \otimes \rho_2(g) \cong \bigoplus_{\rho \in \mathcal{R}(G)} \rho^{\oplus c_{\rho_1, \rho_2, \rho}}, \tag{1}$$

where $\rho^{\oplus m}$ just means the direct sum of $m$ copies of $\rho$, $\bigoplus_{k=1}^m \rho$, and $\mathcal{R}(G)$ is the set of all irreducible representations of $G$; we also note the simple relation $\chi_{\rho_1 \oplus \rho_2} = \chi_{\rho_1} + \chi_{\rho_2}$. Let $\chi_{1 \otimes 2}$ be the character corresponding to $\rho_1 \otimes \rho_2$. An important observation is that $\chi_{1 \otimes 2}(g) = \chi_1(g)\chi_2(g)$, a relation that may be understood in the context of the tracial identity for matrix tensor products; namely, $\text{tr}(A \otimes B) = \text{tr}(A)\text{tr}(B)$. Thus, we have

$$\chi_1(g)\chi_2(g) = \sum_\rho c_{\rho_1, \rho_2, \rho} \chi_\rho(g). \tag{2}$$

(2) induces a canonical polynomial algebra structure on characters, which we will exploit in our approach. In the literature, the coefficients $c_{\rho_1, \rho_2, \rho}$ are known as the *Clebsch-Gordan coefficients*; they are well understood for many groups (Costa & Fogli, 2012). These coefficients are known to be of fundamental importance in quantum mechanics, where they arise in the consideration of independent quantum systems.

## 4 OUR PARADIGM : NON-LINEAR APPROXIMATION VIA NEURAL NETWORKS

### 4.1 LINEAR VS NON-LINEAR APPROXIMATION

In this work, we focus on the task of effective approximation of class functions with a limited alphabet (i.e., dictionary or collection) of irreducible characters, thereby achieving a kind of non-linear dimension reduction. Peter-Weyl theory guarantees that the set $\chi(G)$ of irreducible characters form an orthonormal basis for the space of class functions on $G$. Therefore, in principle, any class function $f$ can be learned simply by linear regression onto $\chi(G)$. However, $\chi(G)$ is finite if and only if $G$ is finite; so for infinite compact groups (such as SO(3)) or even large finite groups, we

can only aspire to learn $f$ from irreducible characters upto sufficiently high order while maintaining a reasonable computational expense. In natural applications, the class function of interest may be highly irregular (e.g. a finite linear combination of indicators in a classification problem); since characters are generally highly regular (e.g., smooth) functions, we would require a very large number of characters to obtain effective linear approximation of irregular functions.

## 4.2 NON-LINEARITY VIA NEURAL NETS

The cornerstone of our approach is to exploit the non-linear, polynomial tensor algebra structure of the group characters in order to amplify and augment the expressive power of a small alphabet of irreducible characters (c.f., Section 3). In implementing our programme, a key role is played by neural networks, which allow us to obtain effective non-linear approximations of class functions in an efficient manner. Apart from their well-known effectiveness in tackling non-linearities, we do not need to a-priori specify explicit polynomial expressions of higher order characters in terms of those in the alphabet; such expressions can be complicated to compute explicitly.

## 4.3 A NEURAL NETWORK BASED CONJUGATION INVARIANT LEARNING PROTOCOL WITH A SMALL ALPHABET

We delineate below the main steps of our procedure to approximate a class function $f$ on a group $G$.

---

**Input.** A small subset of irreducible characters $\{\chi_1, \chi_2, \ldots, \chi_w\} \subset \mathcal{R}(G)$ as alphabet; input data in the form of pairs $\{(g_i, f(g_i)) : g_i \in G; 1 \leq i \leq m\}$.

**Step 1.** For each $1 \leq i \leq m$, compute $\mathbf{v}_i = (\chi_1(g_i), \chi_2(g_i), \ldots, \chi_w(g_i))$.

**Step 2.** Use $\{(\mathbf{v}_i, f(g_i)) : 1 \leq i \leq m\}$ as the training dataset to train a neural network with width $w$ for the initial layer, and width 1 for the final layer.

**Step 3.** For any new input argument $g \in G$, we compute the prediction $\hat{f}(g)$ as follows:

**Step 3a.** Compute $\mathbf{v}_g = (\chi_1(g), \chi_2(g), \ldots, \chi_w(g))$.

**Step 3b.** Perform forward propagation in the neural network output in Step 2 above, with input $\mathbf{v}_g$.

**Step 3c.** Output the result of Step 3b above as the estimate $\hat{f}(g)$.

---

## 4.4 A GLOBAL COORDINATE SYSTEM AND NON-LINEAR DIMENSION REDUCTION

Our approach, in essence leads to a global co-ordinate system for learning conjugation invariant functions on groups (see Sec. B in appendix for detailed discussion). The global co-ordinates of a point $g \in G$ would be given by $(\chi_1(g), \ldots, \chi_w(g))$, where $\{\chi_i\}_{i=1}^w$ is our small alphabet of characters. This stands in contrast to local co-ordinate systems on manifolds or Lie group, which are based on local charts and therefore are effective only in small neighbourhoods; for discrete groups, even such local charts are not available. Further, our alphabet can effectively approximate general class functions via non-linear combinations, thereby according us a low dimensional non-linear co-ordinate system to approximate the infinite dimensional space of class functions. From this perspective, our limited alphabet of characters may be seen as a non-linear dimension reduction tool for class functions.

## 5 THE EXPRESSIVE POWER OF SMALL ALPHABETS AND NON-LINEARITY

In this section, we explore the expressive power of small alphabets of characters for some significant groups. We demonstrate that non-linear, polynomial functions of a very small collection of irreducible characters in groups such as SO(3) and SU(3) are sufficient to generate all characters in these groups, thereby leading to an approximation of arbitrary class functions. This provides a strong theoretical basis to substantiate our approach of learning arbitrary class functions using deep neural networks with a small set of low order irreducible characters as global coordinates. Although the results presented in this section focus on SO(3) and SU(3), we believe that similar analysis can be extended to other groups of practical significance. Such groups often admit a detailed representation theory in general, and an explicit Clebsch-Gordan tensorial decomposition in particular, stemming from their importance in physics and applied mathematics. We refer the reader to Sec. A in the appendix for

discussion on the symmetric group $S_n$. For computational techniques for general groups, see Unger (2006); Babai & Rónyai (1990) in the discrete and Blaha (1969); Green (1971) in the continuum setting. The degree of the polynomial in a small alphabet that is needed to express a higher order character may be taken as a measure of the expressive power of the alphabet, with a low degree signifying high expressive power. Below, we demonstrate that for groups of fundamental interest in applications, we are able to obtain high expressive power with a very small alphabet of irreducible characters.

## 5.1 EXPRESSIVE POWER AND CLEBSCH-GORDAN STRUCTURE FOR SO(3).

The irreducible representations of SO(3), the 3D rotation group, may be conveniently labelled by $\{\rho_j \mid j \in \mathbb{Z}_{\geq 0}\}$, where each $\rho_j$ is of order $2j + 1$ ($\rho_0$ is the one-dimensional trivial representation). Let $\chi_j$ be the corresponding irreducible character. The Clebsch-Gordan decomposition for SO(3) turns out to be very explicit. For $l \geq m$, we have

$$\rho_l \otimes \rho_m \cong \rho_{l+m} \oplus \rho_{l+m-1} \oplus \cdots \oplus \rho_{l-m}.$$

In terms of characters, we have for $l \geq m$

$$\chi_l \cdot \chi_m = \chi_{l+m} + \chi_{l+m-1} + \cdots + \chi_{l-m}. \tag{3}$$

**Theorem 5.1.** *Fix any $l \geq 0$. Then, $\chi_l \in \mathbb{Z}[\chi_1]$. Moreover, $\chi_l$ is a polynomial of degree $l$.*

*Proof.* Setting $m = 1$ in (3) yields

$$\chi_{l+1} = \chi_l \cdot \chi_1 - \chi_l - \chi_{l-1}, \quad l \geq 1. \tag{4}$$

We will prove this result by induction on $l$.

Base case $l \leq 1$ : $\chi_0$ is the trivial character, which is the constant function $\chi_0(g) = 1 \ \forall g \in SO(3)$, so it is a polynomial of degree 0. We of course have that $\chi_1$ is a polynomial of degree 1 in $\mathbb{Z}[\chi_1]$.

Suppose that for every $0 \leq j \leq l$, $\chi_j$ lies in $\mathbb{Z}[\chi_1]$ and that it is a polynomial of degree $j$. We want to prove that $\chi_{l+1}$ lies in $\mathbb{Z}[\chi_1]$ and that it is a polynomial of degree $l + 1$.

By (4), we have $\chi_{l+1} = \chi_l \cdot \chi_1 - \chi_l - \chi_{l-1}$. By our inductive hypothesis, the right-hand side is evidently a polynomial of degree $l + 1$ in $\mathbb{Z}[\chi_1]$. □

## 5.2 EXPRESSIVE POWER AND CLEBSCH-GORDAN STRUCTURE FOR SU(3).

SU(3), the group of $3 \times 3$ unitary matrices with determinant 1, is of key interest in the celebrated Yang-Mills theory and related areas of physics, and has seen a recent surge of activity from a machine learning perspective (Matsumoto et al., 2021; Anderson et al., 2020). The irreducible representations of SU(3) may be succinctly labelled as $\{D(p, q) \mid p, q \in \mathbb{Z}_{\geq 0}\}$; in physical terms, $p$ is the number of quarks and $q$ is the number of antiquarks ($D(0, 0)$ is the trivial representation) (Hall, 2015). $D(p, q)$ is known to have order $\frac{1}{2}(p + 1)(q + 1)(p + q + 2)$. There are many algorithms for computing the Clebsch-Gordan coefficients for SU(3) in general; however, they are rather complicated in nature. We list three simple recursive relations which are sufficient for our study of expressive power.

$$D(p, 0) \otimes D(0, q) \cong D(p, q) \oplus [D(p - 1, 0) \otimes D(0, q - 1)], \quad p, q > 0.$$

$$D(p, 0) \otimes D(1, 0) \cong D(p + 1, 0) \oplus D(p - 1, 1), \quad p > 0.$$

$$D(0, q) \otimes D(0, 1) \cong D(0, q + 1) \oplus D(1, q - 1), \quad q > 0.$$

In terms of irreducible characters, we have respectively for $p, q > 0; p > 0$ and $q > 0$ :

$$\chi^{p,q} = \chi^{p,0}\chi^{0,q} - \chi^{p-1,0}\chi^{0,q-1}; \quad \chi^{p+1,0} = \chi^{p,0}\chi^{1,0} - \chi^{p-1,1}; \quad \chi^{0,q+1} = \chi^{0,q}\chi^{0,1} - \chi^{1,q-1} \tag{5}$$

We now use these recursive relations to prove by induction on $p + q$ that every irreducible character $\chi^{p,q}$ occurs as a polynomial of the two lowest order (order 3) non-trivial irreducible characters $\chi^{1,0}$ and $\chi^{0,1}$.

**Theorem 5.2.** *Fix any $p, q \in \mathbb{Z}_{\geq 0}$. Then, $\chi^{p,q} \in \mathbb{Z}[\chi^{1,0}, \chi^{0,1}]$. Moreover, $\chi^{p,q}$ is a polynomial of degree at most $p + q$.*

*Proof.* We prove the statement by induction on $N = p + q$.

Base case $N = p + q \leq 1 : \chi^{0,0} = 1, \chi^{1,0}, \chi^{0,1} \in \mathbb{Z}[\chi^{1,0}, \chi^{0,1}]$. There is nothing to show here.

Suppose that $\chi^{r,s} \in \mathbb{Z}[\chi^{1,0}, \chi^{0,1}]$, having degree $\leq r + s$, whenever $r + s \leq N$. We want to prove that $\chi^{p,q} \in \mathbb{Z}[\chi^{1,0}, \chi^{0,1}]$, with degree $\leq p + q$, whenever $p + q = N + 1$.

Case 1: $p + q = N + 1, p, q > 0$. By (5), we have $\chi^{p,q} = \chi^{p,0}\chi^{0,q} - \chi^{p-1,0}\chi^{0,q-1}$. The right-hand side is a polynomial of the form $\chi^{r,s}$ such that $r + s \leq N$, therefore $\chi^{p,q} \in \mathbb{Z}[\chi^{1,0}, \chi^{0,1}]$ as required.

Case 2: $p = N+1, q = 0$. By (5), we have $\chi^{N+1,0} = \chi^{N,0}\chi^{1,0} - \chi^{N-1,1}$. Once again, the right-hand side is a polynomial of the form $\chi^{r,s}$ such that $r + s \leq N$, therefore $\chi^{N+1,0} \in \mathbb{Z}[\chi^{1,0}, \chi^{0,1}]$.

Case 3: $p = 0, q = N + 1$. By (5), we have $\chi^{0,N+1} = \chi^{0,N}\chi^{0,1} - \chi^{1,N-1}$. Once again, the right-hand side is a polynomial of $\chi^{r,s}$ such that $r + s \leq N$, therefore $\chi^{0,N+1} \in \mathbb{Z}[\chi^{1,0}, \chi^{0,1}]$. $\quad\square$

## 6 EXPERIMENTS

In this section, we provide a brief description of demonstrative numerical experiments to show the feasibility and efficacy of our learning paradigm in several distinct contexts, . The reader may refer to the appendix Sec. F for a more detailed description and additional experiments. We used TensorFlow (Apache License 2.0) and Keras (MIT License) to run our experiments Abadi et al. (2015); Chollet et al. (2015) on an Intel i7-5500U chip.

The group SO(3) is the standard rotation group on the Euclidean space $\mathbb{R}^3$, significant in many applications such as imaging. Let $\chi_j$ be the irreducible character of SO(3) as mentioned in Sec. 5.1. We use a fully connected neural network with 3 hidden layers (**ConjInv**) with the ReLU activation function at every layer except the last and input alphabet $\{\chi_0, \chi_1, \ldots, \chi_{10}\}$ (see Sec. 4.3) to learn class functions $f$ (specified below) on SO(3). For each experiment, 24000 points $g = (\alpha, \beta, \gamma) \in [0, 2\pi] \times [0, \pi] \times [0, 2\pi]$ on SO(3) are picked uniformly at random to generate our data of the form $(g, f(\beta))$ (note that class functions on SO(3) only depend on $\beta$). We split our data into train, validation and test sets with 20000, 2000 and 2000 points respectively. Throughout this section, we minimize the squared error loss (MSE) using Adam Kingma & Ba (2017). Hyperparameters are tuned according to the validation set and the final performance is measured on the test set.

We benchmark our results against a fully connected neural network with 6 hidden layers (**DeepEuler**) that takes the group elements $g = (\alpha, \beta, \gamma) \in$ SO(3) directly as inputs as well as a linear regression model (**Regression**) with input alphabet $\{\chi_0, \chi_1, \ldots, \chi_{32}\}$. The Euler angles fully describe the group elements in SO(3), and thus a comparison of **DeepEuler** with **ConjInv** provides a concrete measure of the effectiveness of an alphabet of irreducible characters. On the other hand, since characters form a linear basis of the space of conjugation-invariant functions, regression on characters will provide nearly exact learning if the number of characters is high, which suggests such regression as another benchmark to compare against. Our experiments demonstrate that **conjugation invariant learning with group characters outperforms both DeepEuler and Regression.**

**Learning continuous class functions on SO(3).** We randomly generate 100 class functions $f$ as Gaussian linear combinations of $\chi_{50}, \chi_{51}, \ldots, \chi_{59}$ to be learned by **ConjInv**, **DeepEuler** and **Regression** as described above. The three models are compared based on their log-loss on the test set. See Fig. 1.

**Learning discontinuous class functions on SO(3).** We randomly generate 100 discontinuous class functions $f$ by taking Gaussian linear combinations of some binary valued indicator functions $\mathbf{1}_0, \mathbf{1}_1, \ldots, \mathbf{1}_9$, where $\mathbf{1}_k(\alpha, \beta, \gamma) = \begin{cases} 1 & \text{if } 0.1\pi k \leq \beta \leq 0.1\pi k + 0.12\pi, \\ 0 & \text{otherwise.} \end{cases}$ These functions are learned by **ConjInv**, **DeepEuler** and **Regression** as described above. The three models are compared based on their log-loss on the test set. See Fig. 2.

**Learning class functions on $S_n$.** This is an important model of a highly non-commutative *discrete group* that arises in various applications. For the detailed experiment, we refer the reader to Section F.1 the appendix.

**Learning discontinuous class functions on SU(3).** This is a crucial model of a Lie group that has significant physical applications, including in particular in quantum chromodynamics Halzen &

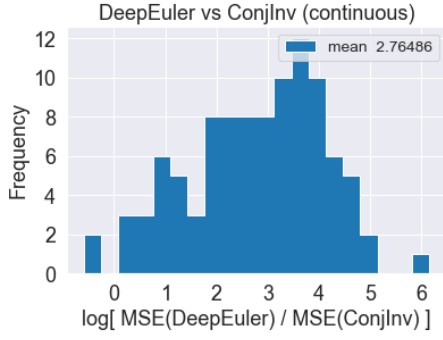 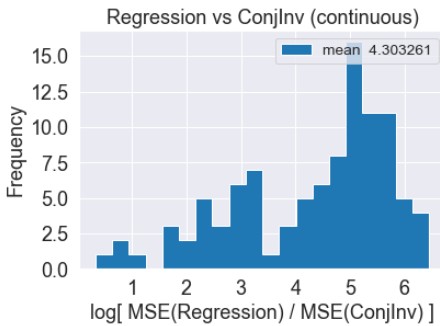

Figure 1

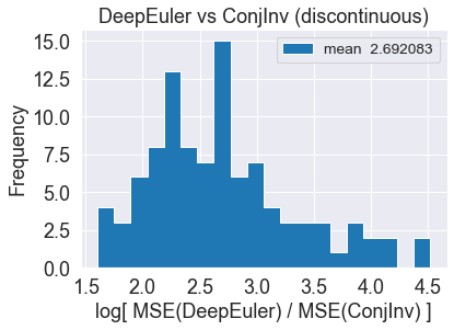 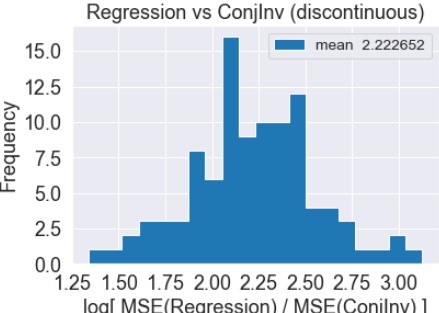

Figure 2

Martin (2008); additional computational challenges result from the matrix entries being complex numbers. We refer the reader to the Section F.4 in the appendix.

## 7 DISCUSSION

We investigate conjugation invariant functions on groups, and propose an effective learning paradigm for such functions via non-linear functions of a small alphabet of characters. We substantiate our approach via theoretical analysis on several groups significant in applications, and via synthetic experiments using neural networks to efficiently capture the non-linearities.

To our knowledge, the present submission is the first work to explore the problem of learning conjugation invariant functions. Another key aspect of our work is to propose the extensive use of group characters, particularly their non-linear polynomial functionals, as a tool in machine learning under group action. Further, we set up our approach to be applicable in the context of the actions of any group, whereas many of the popular methods (such as CNNs) are tailored to specific kinds of group actions, or specific application domains. As such, there is very little benchmark to compare against for the problem at hand. We emphasize that, given that this is the first exploration in this area, our experiments are more for demonstrative purposes to provide proof-of-concept for our paradigm; the main contributions of the paper being in the realm of theory and ideas.

The investigations in the present paper call forth natural directions for future work, including experiments with real-life data, a theoretical as well as experimental study of how to optimally choose the size of the input character alphabet in relation to the computational budget available (e.g., depth of the neural net), a theoretical analysis of expressive power that subsumes general classes of group actions, and domain-specific applications such as graph neural networks Maron et al. (2018); Azizian & Lelarge (2020); Keriven & Peyré (2019). We believe that our work will motivate new perspectives within the general ambit of learning under group actions.

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

# A    EXPRESSIVE POWER AND CLEBSCH-GORDAN STRUCTURE FOR SYMMETRIC GROUPS.

Let $S_n$ be the symmetric group on $n$ letters. Two permutations in $S_n$ are conjugate if and only if they have the same cycle shape.

**Definition.** We say that an $m$-tuple $(\lambda_1, \lambda_2, \dots, \lambda_m)$ is a *partition* of $n$ if $\lambda_i \in \mathbb{Z}_{>0}$, $\lambda_{i+1} \leq \lambda_i$ and $\sum_{i=1}^{m} \lambda_i = n$. The number of parts of $\lambda$ is denoted by $l(\lambda) = m$.

We denote the set of all partitions of $n$ by $\mathcal{P}(n)$. A natural ordering of $\mathcal{P}(n)$ is the *lexicographic order*.

**Definition.** If $\lambda$ and $\mu$ are partitions of $n$, we write $\lambda > \mu$ if and only if the least $j$ for which $\lambda_j \neq \mu_j$ satisfies $\lambda_j > \mu_j$.

Cycle shapes may be canonically represented by partitions of $n$. It is a well known fact that for any finite group $G$, the set of irreducible representations of $G$ over $\mathbb{C}$ is in bijection with the conjugacy classes of $G$ (in particular, character tables for finite groups are square). Therefore, it is not surprising that the irreducible representations of $S_n$ may be conveniently labelled by the partitions of $n$. For each $\lambda \in \mathcal{P}(n)$, there is an associated irreducible representation $V^\lambda$ (known as Specht modules in the literature) of $S_n$. Some examples are the one-dimensional trivial representation $V^{(n)}$ and the one-dimensional sign representation $V^{(1^n)}$, where $(1^n)$ is a shorthand for the partition $(1, 1, \dots, 1) \in \mathcal{P}(n)$. For a more detailed description of the Specht modules, the reader may refer to Gordan James's classical textbook James (2006).

**Definition.** Let $\lambda \in \mathcal{P}(n)$. The *Young diagram* of $\lambda$ is defined to be the set

$$[\lambda] = \{(a, b) \in \mathbb{Z}_{>0} \times \mathbb{Z}_{>0} \mid 1 \leq a \leq l(\lambda), 1 \leq b \leq \lambda_a\}.$$

The elements of $[\lambda]$ are called the *nodes* of $\lambda$.

**Definition.** The *conjugate* of a partition $\lambda$ is the partition $\lambda' = (\lambda'_1, \lambda'_2, \dots)$ such that $\lambda'_i = \#\{j \geq 1 \mid \lambda_j \geq i\}$.

In other words, the Young diagram $[\lambda']$ may be obtained by swapping the rows and columns of $[\lambda]$:

$$[\lambda'] = \{(b, a) \in \mathbb{Z}_{>0} \times \mathbb{Z}_{>0} \mid (a, b) \in [\lambda]\}.$$

In particular, we note that $(\lambda')' = \lambda$. This induces a pairing of partitions in $\mathcal{P}(n)$.

The Young diagram of a partition $\lambda$ may be represented pictorially by a diagram. For each $(i, j) \in [\lambda]$, we insert a box into the $i^{th}$ row and $j^{th}$ column of the diagram.

*Example.* Suppose that $\lambda = (7, 6, 5, 5, 1) \in \mathcal{P}(24)$ so that $l(\lambda) = 5$. $[\lambda]$ :

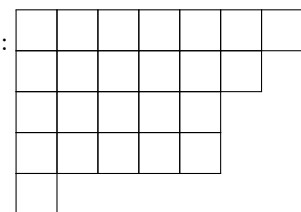

$\lambda' = (5, 4^4, 2, 1)$. $[\lambda']$ :

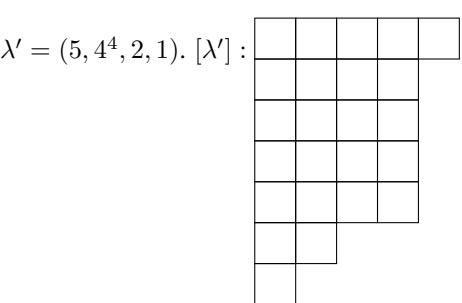

The Clebsch-Gordan coefficients for $S_n$ are also known as the Kronecker coefficients in the literature. A fundamental problem in the representation theory of $S_n$ is to understand the Kronecker coefficients. Apart from some special cases Bowman et al. (2015); Tewari (2015); Hamermesh (2012) such as when the partitions have only two parts or are hook-shaped, there are more questions than answers in general.

A fundamental yet simple method of constructing representations of $S_n$ is by taking an existing representation and tensoring it with the one-dimensional sign representation $V^{(1^n)}$. Doing this for the irreducible Specht module $V^\lambda$ produces its conjugate Specht module $V^{\lambda'}$:

$$V^\lambda \otimes V^{(1^n)} \cong V^{\lambda'} \qquad \forall \lambda \in \mathcal{P}(n).$$

Let $\chi^\lambda$ be the irreducible character corresponding to $V^\lambda$. In terms of characters, the above result is simply

$$\chi^\lambda \chi^{(1^n)} = \chi^{\lambda'}. \tag{1}$$

Let $\mathcal{R}(G)$ denote the set of irreducible characters of a group $G$. This means that we may pick a single representative character $\chi^\lambda$ for every pair of characters $(\chi^\lambda, \chi^{\lambda'})$ when curating an alphabet $X$ to generate $\mathcal{R}(S_n)$ as a polynomial in $\mathbb{R}[X]$. In other words, the size of the minimal generating alphabet $X$ is at most about half that of $\mathcal{R}(S_n)$. We believe that in most cases, the minimal generating alphabet size will be even smaller. However, due to the difficulty of working with Kronecker coefficients, a general result of this form is beyond the scope of this paper and may be discussed in a future work. We demonstrate this with a small example using only one more basic fact Hamermesh (2012) about Kronecker coefficients. For any $n \in \mathbb{Z}_{>0}$ and $\lambda \in \mathcal{P}(n)$,

$$\chi^\lambda \chi^{(n-1,1)} = (\#\{\lambda_i\} - 1)\chi^\lambda + \sum_\mu \chi^\mu, \tag{2}$$

where the sum is over all partitions $\mu$ whose Young diagram $[\mu]$ is obtained from the Young diagram $[\lambda]$ by removing a node and adding another node, and $\#\{\lambda_i\} - 1$ is one less than the number of different row lengths of $[\lambda]$.

*Example.* Let $n = 6$. We list all 11 partitions of 6 in the following table:

| $\lambda$ | $(6)$ | $(5,1)$ | $(4,2)$ | $(4,1,1)$ | $(3,3)$ | $(3,2,1)$ |
|---|---|---|---|---|---|---|
| $\lambda'$ | $(1^6)$ | $(2,1,1,1,1)$ | $(2,2,1,1)$ | $(3,1,1,1)$ | $(2,2,2)$ | $(3,2,1)$ |

Recall that $\chi^{(6)}$ is the trivial character that takes constant value 1. By (1), we have $\mathbb{R}[\mathcal{R}(S_6)] = \mathbb{R}[\chi^{(5,1)}, \chi^{(4,2)}, \chi^{(4,1,1)}, \chi^{(3,3)}, \chi^{(3,2,1)}, \chi^{(1^6)}]$. We show that in fact,

$$\mathbb{R}[\mathcal{R}(S_6)] = A := \mathbb{R}[\chi^{(5,1)}, \chi^{(4,2)}, \chi^{(1^6)}].$$

It suffices to show that $\chi^{(4,1,1)}$, $\chi^{(3,3)}$ and $\chi^{(3,2,1)} \in A$. By (2), we have $\chi^{(5,1)}\chi^{(5,1)} = 1 + \chi^{(5,1)} + \chi^{(4,2)} + \chi^{(4,1,1)}$, thus $\chi^{(4,1,1)} \in A$.

Using (2) again, we obtain

$$\chi^{(5,1)}\chi^{(4,2)} = \chi^{(5,1)} + \chi^{(4,2)} + \chi^{(4,1,1)} + \chi^{(3,3)} + \chi^{(3,2,1)} \text{ and}$$

$$\chi^{(5,1)}\chi^{(4,1,1)} = \chi^{(5,1)} + \chi^{(4,2)} + \chi^{(4,1,1)} + \chi^{(3,2,1)} + \chi^{(3,1,1,1)}.$$

Hence, $\chi^{(3,3)} + \chi^{(3,2,1)} \in A$ and $\chi^{(3,2,1)} + \chi^{(3,1,1,1)} \in A$. Since $(3,1,1,1) = (4,1,1)'$, we have $\chi^{(3,1,1,1)} = \chi^{(4,1,1)}\chi^{(1^6)} \in A$ by (1).

Therefore, $\chi^{(3,2,1)} \in A$ and $\chi^{(3,3)} \in A$.

This completes the proof that $\mathbb{R}[\mathcal{R}(S_6)] = \mathbb{R}[\chi^{(5,1)}, \chi^{(4,2)}, \chi^{(1^6)}]$.

*Remark.* Similarly, one can show using (1) and (2) that $\mathbb{R}[\mathcal{R}(S_7)] = \mathbb{R}[\chi^{(6,1)}, \chi^{(5,2)}, \chi^{(1^7)}]$. Note that $|\mathcal{P}(7)| = 15$.

## B    GLOBAL COORDINATES VIA IRREDUCIBLE CHARACTERS

In the main text, Sec. 4.4, we indicated that our approach, in essence leads to a global co-ordinate system for learning conjugation invariant functions on groups. To be precise, the global co-ordinates of a point $g \in G$ would be given by $(\chi_1(g), \ldots, \chi_w(g))$, where $\{\chi_i\}_{i=1}^w$ is our small alphabet of characters. Herein, we discuss further details and geometric implications of this, which we needed to abbreviate in the main text. We emphasize that, these global co-ordinates are with a view to learning conjugation invariant functions on the group $G$. We recall here that conjugation invariant functions are constant on the conjugacy classes, which are the orbits of the conjugation action of $G$ on itself. For optimality of representation, it is natural that a coordinate system for efficient learning of class functions would be constant on the conjugacy classes of the group. This is satisfied by our character-induced coordinate system $(\chi_1(\cdot), \ldots, \chi_w(\cdot))$, which is constant on the conjugacy classes of $G$. Thus, in precise terms, these global coordinates parametrize the space of conjugacy classes of the group, which is minimally sufficient for describing class functions on $G$. On a Lie group (such as SO(n) or SU(n)), the conjugation action of the group is a smooth action, i.e. the map $\tau(g,x) = gxg^{-1}$ is a smooth map taking $G \times G \mapsto G$. The decomposition of the group into conjugacy classes corresponds to a *foliation* of $G$ (viewed as a manifold, c.f. (Lawson Jr, 1974; Walschap, 2012)) into submanifolds (called the *leaves* of the foliation) corresponding to the conjugacy classes, on which any class functions is constant. Thus, from a geometric point of view, the coordinates induced by the irreducible characters provide a global parametrization for the leaves of the foliation of $G$ into its conjugacy classes.

## C    LIFTING FUNCTIONS ON HOMOGENEOUS SPACES TO GROUPS

In this section, we elaborate on the ideas articulated in the main text, pertaining to the lifting of functions from homogeneous spaces to groups, thereby substantiating our focus on studying functions defined on groups.

In our considerations, we mostly focus on the scenario where the conjugation invariant function is defined on the group, and the group is envisaged to be acting on itself. In many situations of interest in ML, the group of symmetries actually happen to act on a set of objects that are directly not elements of the group itself. Such a set on which the group acts is sometimes referred to as a homogeneous space of the group. However, we demonstrate that our setup is easily extended to cover such situations.

In the common situation where the group $G$ acts transitively on the homogeneous space $X$ (i.e. any pair of elements in $X$ can be mapped to each other by appropriate group elements, e.g. the Euclidean rotation group acting on the sphere), it is known that the homogeneous space $X$ can be identified with a quotient of the group $G$ by a certain subgroup (the stabilizer subgroup of any given point in $X$). This is entailed by the celebrated Orbit-Stabilizer Theorem. In fact, this essentially identifies $X$ as the coset space of such a stabilizing subgroup of $G$.

In the context of our work, the upshot of this is that, any function $f : X \mapsto \mathbb{R}$ can be 'lifted' to a function on $G$, which is to say that there is a function $\tilde{f} : G \mapsto \mathbb{R}$ and a surjective quotient map $\pi : G \mapsto \mathbb{R}$ such that $\tilde{f}(g) = f(\pi(g)) \forall g \in G$. Since $\pi$ is surjective, the functional value of $f$ at

any point $x \in X$ can be obtained from $\tilde{f}$ in accordance with the above formula. This procedure is completely canonical, and can be easily carried out explicitly in a general setting. The lifting construction is succinctly summarized in the following commutative diagram:

$$
\begin{array}{ccc}
 & & G \\
 & \pi \downarrow & \diagdown \, \tilde{f} \\
X & \xrightarrow{\ f\ } & \mathbb{R}
\end{array}
$$

A simple yet informative example of the above scenario is accorded by the action of the permutation group $S_n$ on the labels $[n]$ of a set $X$ of $n$ objects. If we fix a specific object, then its label is stabilized or left unchanged under this action by a Stab subgroup of $S_n$ that permutes only the labels of the remaining $n-1$ objects, and is thus isomorphic to $S_{n-1}$. The set $X$, by the preceding discussion, is therefore in bijective correspondence with the coset space $S_n/\mathrm{Stab}$; notice that since $\mathrm{Stab} \cong S_{n-1}$, we have $|S_n/\mathrm{Stab}| = |S_n/S_{n-1}| = n$. The cosets of the subgroup Stab form a disjoint partition of $S_n$: in other words, $S_n = \bigcup \mathrm{Stab}_i$, where $\mathrm{Stab}_i$ are the disjoint cosets of Stab inside $S_n$. Any function $f : X \mapsto \mathbb{R}$ can be lifted to a function $\tilde{f}$ according to the above recipe as follows. Any $\sigma \in S_n$ belongs to a unique coset $\mathrm{Stab}_{i_\sigma}$ above. On the other hand, each coset $\mathrm{Stab}_i$ maps to a unique element $x_i \in X$ under the Orbit-Stabilizer paradigm. We construct $\tilde{f}$ by setting $\tilde{f}(\sigma) = f(x_{i_\sigma})$. It is easy to verify that $\tilde{f}$ indeed satisfies the requirements of the commutative diagram above, thereby verifying that it is a legitimate lift of $f$ from $X$ to $S_n$, as desired.

We conclude this section with a theoretical analysis of how the group action on a homogeneous space $X$ decomposes $X$ into a disjoint union of commutator orbits.

**Lemma C.1.** *Let the group $G$ act transitively on the homogeneous space $X$. Let $f : X \mapsto \mathbb{R}$, with the lift $\tilde{f} : G \mapsto \mathbb{R}$ such that $\tilde{f} = f \circ \pi$, where $\pi : G \mapsto X$ is the canonical projection map. Then $\tilde{f}$ being conjugation invariant under $G$ is equivalent to $f$ being invariant under the induced action of $[G, G]$ on $X$, where $[G, G]$ is the commutator subgroup of $G$ generated by $\{[g, h] = ghg^{-1}h^{-1} : g, h \in G\}$.*

*Proof.* Fix $x_0 \in X$. We define $\tilde{f}(g) = f(g \cdot x_0)$ for $g \in G$. $\tilde{f}$ is conjugation invariant if $\tilde{f}(ghg^{-1}) = \tilde{f}(h) \forall g, h \in G$. In terms of $f$, we have $f(ghg^{-1} \cdot x_0) = f(h \cdot x0) \forall g, h \in G$. Writing $x = h \cdot x_0$, we may write $f(ghg^{-1}h^{-1}x) = f(x) \forall g, h \in G$, that is, $f([g, h] \cdot x) = f(x) \ \forall g, h \in G$, where $[g, h] = ghg^{-1}h^{-1}$ is the commutator of $g$ and $h$. Thus, $\tilde{f}$ is conjugation invariant implies that $f$ is $[G, G]$ invariant. It is easy to see that we can reverse this argument to obtain the converse implication. $\square$

To summarize, the full group $G$ acts transitively on the homogeneous space $X$, enabling a lift of the function $f$ to a function $\tilde{f}$ on $G$. This function being conjugation invariant on $G$ amounts to the function $f$ being invariant to the induced action of the commutator subgroup $[G, G]$ on $X$. We may view this as the homogeneous space $X$ being partitioned into $[G, G]$-orbits, whereby $f$ is constant on each $[G, G]$-orbit.

For any group $G$, the commutator subgroup $[G, G]$ is the smallest normal subgroup $N$ of $G$ such that $G/N$ is abelian. In some sense, $[G, G]$ captures all the non-abelian attributes of $G$.

Commutators are well-studied objects in group theory. For example, it is known that the commutator of the symmetric group $S_n$ is the alternating group $A_n$. For the dihedral group $D_{2n}$, which is generated by a reflection $x$ and a rotation $y$ satisfying the relations $x^2 = y^n = 1, xyx = y^{-1}$, the commutator $[G, G]$ is the cyclic subgroup generated by $y^2$.

## D   COMPACT VS NON-COMPACT GROUPS

We emphasize that, in the present work, we focus on the case of compact groups; finite groups trivially belonging to that category in particular. Compact groups are, in a sense, a natural point of departure for investigations such as ours for an array of reasons. On one hand, compact groups (such as rotations or permutations) provide some of the most fundamental models for learning under non-abelian group actions. On the other hand, compact groups allow for a very concrete and well-understood representation theory and character theory, as outlined in the main text. However, the action of non-compact groups does arise as symmetries in many interesting applications. Of course,

for locally compact abelian groups, we have an elaborate theory of commutative Fourier analysis on groups, as discussed in the main text (also c.f. (Rudin, 1962)); a ready example being provided by the commutative action of the group of real numbers (or that of $\mathbb{R}^d$ in general) in many natural learning set ups. A natural application of locally compact non-abelian groups is accorded by the group of Euclidean isometries (denoted E(n) or ISO(n)), generated by translations and rotations acting on Euclidean space. In more advanced physical applications, depending on the energy scale and relativistic effects involved, it is also of interest to investigate the action of the Galilean group or the Poincare group (c.f. (Arnol'd, 2013; Nadjafikhah & Forough, 2007; Tung, 1985)). While the representation theory and the character theory of general classes of non-compact non-abelian groups poses a formidable mathematical challenge, the investigation of specific groups relevant to applications opens a natural direction for future work.

# E  GROUP REPRESENTATIONS, IRREDUCIBLE CHARACTERS AND FOURIER ANALYSIS

## E.1  GROUP REPRESENTATIONS, IRREDUCIBLE CHARACTERS AND PETER-WEYL THEORY

We provide here the very broad contours of only a few aspects of the deep theory of group representations Simon (1996), focussing on ingredients that would be germane to our approach. We will consider $\rho$ to be a matrix representation of $G$. That is, $\rho : G \to \mathbb{C}^{d_\rho \times d_\rho}$; $\rho(xy) = \rho(x)\rho(y)$ for any $x, y \in G$; and $\rho(e) = I$, where $e$ is the identity element of $G$ and $I$ is the identity matrix. The *character* $\chi_\rho$ of $\rho$ is the function $\chi : G \to \mathbb{C}$ defined by $\chi(x) = \text{tr}(\rho(x))$. We call $d_\rho$ the *order* of $\rho$ or $\chi_\rho$. We say that $\rho$ is *reducible* if it decomposes into a direct sum of smaller representations as

$$\rho(x) = Q^{-1}(\rho_1(x) \oplus \rho_2(x))Q = Q^{-1} \left( \begin{array}{c|c} \rho_1(x) & 0 \\ \hline 0 & \rho_2(x) \end{array} \right) Q \quad \forall x \in G \text{ for some invertible matrix}$$

$Q \in \mathbb{C}^{d_\rho \times d_\rho}$ which is independent of $x$. Otherwise, we say that $\rho$ is *irreducible*. If $\rho$ is an irreducible representation, we say that $\chi_\rho$ is an *irreducible character*. Since $\text{tr}(T^{-1}AT) = \text{tr}(A)$, it is evident that $\chi_\rho$ is a class function. By the celebrated Peter-Weyl theorem (Simon, 1996), the set of irreducible characters form an orthonornal basis for the space of square integrable class functions.

## E.2  REPRESENTATION THEORY AND NON-COMMUTATIVE FOURIER ANALYSIS.

Classical Fourier analysis, in its simplest avatar, entails that given a function $f : \mathcal{S}^1 \mapsto \mathbb{C}$, we may decompose $f$ as $f(x) = \sum_{k=-\infty}^{\infty} \hat{f}(k)e^{ikx}$, with the Fourier coefficients being defined as $\hat{f}(k) = \frac{1}{2\pi} \int_0^{2\pi} f(x)e^{-ikx}dx$. Representation theory allows us to undertake Fourier analysis on groups. If $f : G \mapsto \mathbb{C}$ (where $G$ is a locally compact abelian group with Haar measure $\mu$), we may decompose $f$ as $f(x) = \int_{\hat{G}} \hat{f}(\chi)\chi(x^{-1})d\hat{\mu}(\chi)$, with $\hat{f}(\chi) = \int_G f(x)\chi(x)d\mu(x)$; here where $\chi$ ranges over the the space of irreducible characters of $G$ (equalling in this case the so-called Pontryagin dual $\hat{G}$). It is known that the irreducible representations of an abelian group are all one-dimensional, so each $\hat{f}(\chi)$ is still a scalar.

If $G$ is *non-abelian* but is *compact*, then the set of irreducible representations, also denoted as $\mathcal{R}(G)$, is known to be countable, and the representing matrices $\rho$ may be taken to be unitary matrices. Given a function $f : G \to \mathbb{C}$, we may decompose $f$ as

$$f(x) = \frac{1}{\mu(G)} \sum_{\rho \in \mathcal{R}(G)} d_\rho \text{tr}[\hat{f}(\rho)\rho(x^{-1})], \quad \text{with} \quad \hat{f}(\rho) = \int_G f(x)\rho(x)d\mu(x). \quad (3)$$

It may be noted in this case that each Fourier coefficient $\hat{f}(\rho)$ is a matrix, leading to the notion of non-commutative Fourier analysis stemming from the fundamentally non-commutative nature of matrix multiplication.

## E.3  PLANCHEREL'S THEOREM AND BAND-LIMITED FUNCTIONS.

In classical Fourier analysis, Plancherel's Theorem tells us that the Fourier transform preserves energy; namely, $\int_{-\infty}^{\infty} |f(x)|^2 dx = \sum_{k=-\infty}^{\infty} |\hat{f}(k)|^2$. Since the sum on the right is finite, $|\hat{f}(k)|^2 \to 0$ as $k \to \infty$. Thus, $f(x)$ is well-approximated by $\sum_{k=k_0}^{k_0} \hat{f}(k)e^{ikx}$ for sufficiently large $k_0$. This is akin

to band-limiting $f$, a device that is extremely useful in signal processing and many other areas of applied mathematics. In the case of compact groups, it is known that the non-commutative Fourier transform $f \mapsto \hat{f}$ defined above satisfies a non-commutative Plancherel's theorem. Indeed,

$$\frac{1}{\mu(G)} \int_G |f(x)|^2 d\mu(x) = \frac{1}{[\mu(G)]^2} \sum_{\rho \in \mathcal{R}(G)} d_\rho \|\hat{f}(\rho)\|_{\text{Frob}}^2. \tag{4}$$

Since the sum on the right of (4) is finite, $\|\hat{f}(\rho)\|_{\text{Frob}}^2 \to 0$ as $d_\rho \to \infty$. Thus, $f(x)$ is well-approximated by $\frac{1}{\mu(G)} \sum_{d_\rho \le k_0} d_\rho \text{tr}[\hat{f}(\rho)\rho(x^{-1})]$ for sufficiently large $k_0$, leading to a notion of non-commutative bandlimiting for functions defined on non-abelian groups.

## F EXPERIMENTS

Using the ideas in Sec. 4.3, we implement several numerical experiments to show how neural networks with small alphabet can be used to approximate various class functions. Our experiments are implemented using TensorFlow (Apache License 2.0), Keras (MIT License) Abadi et al. (2015); Chollet et al. (2015) and SageMath (GPLv3) Stein et al. (2021). Tensorflow and Keras was used to train our models. SageMath was used to compute the irreducible characters of $S_{30}$.

### F.1 LEARNING CLASS FUNCTIONS ON $S_n$

Since the conjugacy classes of $S_n$ are canonically labelled by partitions A, class functions on $S_n$ can be realised as functions on $\mathcal{P}(n)$. Set $n = 30$. There are 5604 partitions of 30. We generate 100 class functions on $S_{30}$ by taking Gaussian linear combinations of 10 irreducible characters $\chi^{(18,4,2,2,1,1,1,1)}, \chi^{(18,4,2,1,1,1,1,1,1)}, \dots, \chi^{(18,3,3,1,1,1,1,1,1)}$; $(18, 4, 2, 2, 1, 1, 1, 1) > (18, 4, 2, 1, 1, 1, 1, 1, 1) > \cdots > (18, 3, 3, 1, 1, 1, 1, 1, 1)$ are the $251^{\text{st}}, 252^{\text{nd}}, \dots, 260^{\text{th}}$ elements of $\mathcal{P}(30)$ in the lexicographic order A. Our training data comprises 1400 ($\sim$25% of $|\mathcal{P}(30)|$) function evaluations $(\lambda, f(\lambda))$, where the partitions $\lambda$ are sampled uniformly at random from $\mathcal{P}(30)$. To approximate this, we train a fully connected neural network with 3 hidden layers of width 180, 120 and 50 respectively. The input layer consists of the 50 irreducible characters $\chi^{(30)}, \chi^{(29,1)}, \dots, \chi^{(22,6,1,1)}, \chi^{(1^{30})}$; these are the irreducible characters corresponding to the first 49 elements of $\mathcal{P}(30)$ in the lexicographic order, on top of the one-dimensional sign character $\chi^{(1^{30})}$. The SELU non-linear activation function and Gaussian kernel initializer is applied throughout. We minimize the squared error loss (MSE) with L2 regularization parameter $\alpha = 0.001$ using AdaMax Kingma & Ba (2017).

As a benchmark, we train a linear regression model with 100 input alphabets $\{\chi^{(30)}, \chi^{(29,1)}, \dots, \chi^{(20,9,1)}, \chi^{(1^{30})}\}$; these are the irreducible characters corresponding to the first 99 elements of $\mathcal{P}(30)$ in the lexicographic order plus the sign character $\chi^{(1^{30})}$. Gaussian kernel initializer is used and we minimize the squared error loss using Adam Kingma & Ba (2017).

Note that we apply the same hyperparameters (chosen to ensure reasonable convergence) for each of the 100 class functions that we train. Our neural network was trained over 200 epochs using a batch size of 100 for each class function. The linear regression model was also trained over 200 epochs using a batch size of 100. Our models are evaluated using 10-fold cross-validation.

In the scatter plot of Figure 1, each point corresponds to the performance of our models on a fixed class function, with the $y$-axis representing the mean MSE over 10 folds and the $x$-axis representing the corresponding standard deviation. Taking the mean MSE over 10 folds as a representative for each scatter point, we obtain the histogram in Figure 1. We display the average of the histogram in its legend. Note that we normalized each of the 100 class functions $f$ so that the mean of squares over the 1400 instances of $f(\lambda)$ is equal to 1.

We find that the performance of the neural network is far superior to the performance of a linear regression model with double the size of the alphabet. The mean MSE over 100 class functions trained for linear regression is 5.10 times higher than that for the neural network.

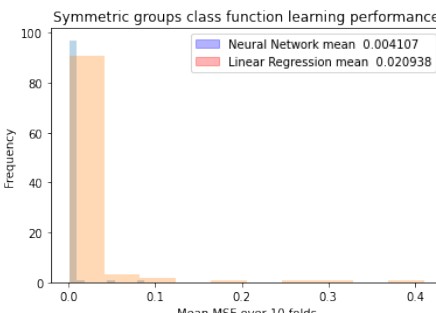 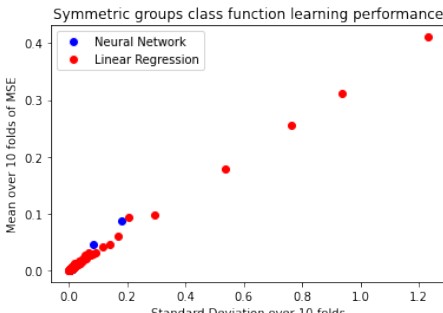

Figure 1

## F.2 LEARNING CONTINUOUS CLASS FUNCTIONS ON SO(3)

This experiment was described briefly in Section 6. We provide the full details here. Let $\chi_j$ be the irreducible character corresponding to the $(2j + 1)$-dimensional irreducible representation $\rho_j$ of SO(3) (see Section 5.1). Then, $\chi_j$ is the real-valued function $\chi_j(\beta) = \frac{\sin((2j+1)\beta/2)}{\sin(\beta/2)}$ for $0 \le \beta \le \pi$ (note that class functions on SO(3) depend only on a single angle $\beta$).

We generate 100 class functions $f$ on SO(3) by taking Gaussian linear combinations of 10 high order irreducible characters $\chi_{50}, \chi_{51}, \ldots, \chi_{59}$. Our data comprises 24000 function evaluations ($g = (\alpha, \beta, \gamma), f(\beta)$), where $g = (\alpha, \beta, \gamma)$ was drawn uniformly at random from $[0, 2\pi] \times [0, \pi] \times [0, 2\pi]$. We split our data into train, validation and test sets with 20000, 2000 and 2000 points respectively. A fully connected neural network (**ConjInv**) with 3 hidden layers of width 100, 50 and 20 respectively is fit to the train set over 180 epochs using a batch size of 1000. The input layer consists of the 11 lowest order irreducible characters $\chi_0, \chi_1, \ldots, \chi_{10}$ of SO(3) (see Sec. 4.3). The ReLU non-linear activation function is used at every layer except the last. Gaussian kernel initializer is applied throughout. We minimize the squared error loss (MSE) with L2 regularization parameter $5 \times 10^{-4}$ using Adam Kingma & Ba (2017) with learning rate $10^{-3}$. The hyperparameters were tuned according to the validation set.

Our first benchmark is a fully connected neural network with 6 hidden layers (**DeepEuler**) of width 150, 120, 80, 60, 30 and 10 respectively. The input layer consists of the three Euler angles $g = (\alpha, \beta, \gamma)$. **DeepEuler** is fit to the train set over 130 epochs using a batch size of 1000. The ReLU non-linear activation function is used at every layer except the last. Gaussian kernel initializer is applied throughout. By tuning hyperparameters according to the validation set, we optimize **DeepEuler** using Adam Kingma & Ba (2017) with learning rate $8 \times 10^{-2}$ subject to L2 regularization parameter $3 \times 10^{-12}$.

As a second benchmark, a linear regression model **Regression** with 33 input alphabets $\chi_0, \chi_1, \ldots, \chi_{32}$ is fit to the train set over 100 epochs using a batch size of 500. Gaussian kernel initializer is used. By tuning hyperparameters according to the validation set, we optimize **Regression** using Adam Kingma & Ba (2017) with learning rate $3 \times 10^{-3}$ and L2 regularization parameter $10^{-15}$.

The three models are evaluated on the test set. We compare their log losses in Figure 2. Positive values in our histogram represent instances when **ConjInv** outperforms its benchmark. Observe that when compared against **Regression**, **ConjInv** has superior performance 100 out of 100 times. On the other hand, **ConjInv** outperforms **DeepEuler** 98 out of 100 times. By taking the average of the histograms, we conclude that on average, the mean squared error loss for **Regression** and **DeepEuler** are $e^{4.30} = 73.7$ and $e^{2.76} = 15.8$ times that of **ConjInv** respectively.

## F.3 LEARNING DISCONTINUOUS CLASS FUNCTIONS ON SO(3)

We provide the full details to the experiment described in Section 6. Let $\chi_j$ be the irreducible character of SO(3) (see Sec. F.2).

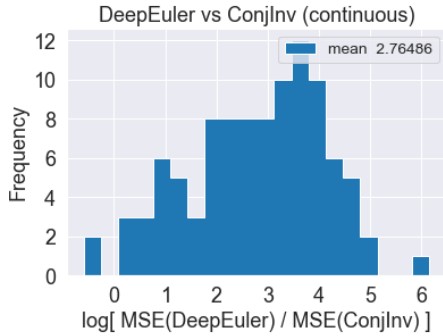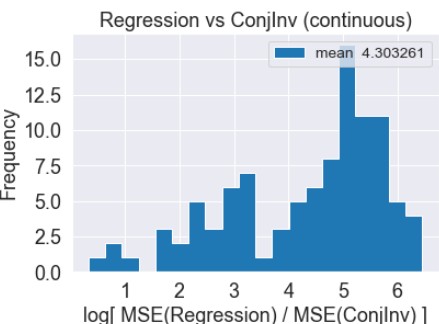

Figure 2

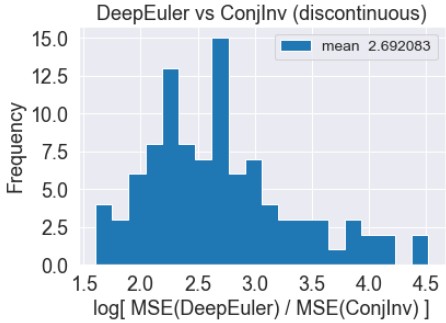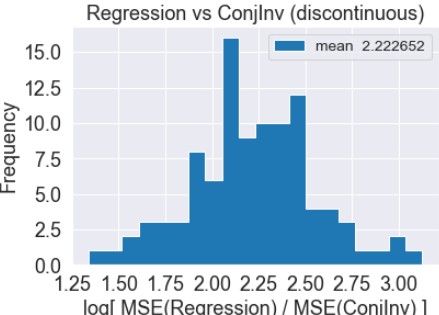

Figure 3

100 discontinuous class functions $f$ on SO(3) are generated by taking Gaussian linear combinations of some binary valued indicator functions $\mathbf{1}_0, \mathbf{1}_1, \dots, \mathbf{1}_9$, where

$$\mathbf{1}_k(\alpha, \beta, \gamma) = \begin{cases} 1 & \text{if } 0.1\pi k \leq \beta \leq 0.1\pi k + 0.12\pi, \\ 0 & \text{otherwise.} \end{cases}$$

Drawing 24000 instances of $g = (\alpha, \beta, \gamma)$ uniformly at random from $[0, 2\pi] \times [0, \pi] \times [0, 2\pi]$, we obtain 24000 datapoints of the form $(g, f(\beta))$. We split our data into train, validation and test sets with 20000, 2000 and 2000 points respectively. We fit **ConjInv** from Sec. F.2 to the train set over 180 epochs using a batch size of 180. By tuning the hyperparameters according to the validation set, we minimize the mean squared error loss with L2 regularization parameter $10^{-15}$ using Adam Kingma & Ba (2017) with learning rate $10^{-3}$.

Just like in Sec. F.2, our first benchmark is **DeepEuler**, which is fit to the train set over 180 epochs using a batch size of 2000. By tuning hyperparameters according to the validation set, we optimize **DeepEuler** using Adam Kingma & Ba (2017) with learning rate $8 \times 10^{-2}$ subject to L2 regularization parameter $10^{-15}$.

We also benchmark against **Regression**, which is fit to the train set over 100 epochs using a batch size of 500. By tuning hyperparameters according to the validation set, we optimize **Regression** using Adam Kingma & Ba (2017) with learning rate $3 \times 10^{-3}$ subject to L2 regularization parameter $10^{-15}$.

The three models are evaluated on the test set. We compare their log losses in Figure 3. Positive values in our histogram represent instances when **ConjInv** outperforms its benchmark. Observe that when compared against **DeepEuler** and **Regression**, **ConjInv** achieves superior performance 100 out of 100 times. By taking the average of the histograms, we conclude that on average, the mean squared error loss for **Regression** and **DeepEuler** are $e^{2.22} = 9.2$ and $e^{2.69} = 14.7$ times that of **ConjInv**.

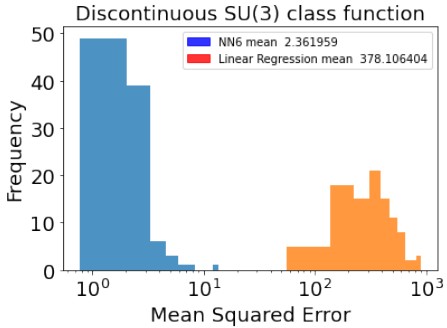

Figure 4

F.4    LEARNING DISCONTINUOUS CLASS FUNCTIONS ON $\mathrm{SU}(3)$

Let $\chi^{p,q}$ be the irreducible character corresponding to the irreducible representation $D(p,q)$ of $\mathrm{SU}(3)$ (see Section 6.2). Then, $\chi^{p,q}$ is the complex-valued function $\chi^{p,q}(\theta,\phi) = e^{i\frac{\theta}{3}(2p+q)}\sum_{k=0}^{p}\sum_{l=0}^{q}e^{-i(k+l)\theta}\left(\frac{\sin((k-l+q+1)\phi/2)}{\sin(\phi/2)}\right)$, for $-2\pi \le \phi \le 2\pi$ and $-3\pi \le \theta \le 3\pi$. Since Keras does not support complex valued data types, we use the Complex-Valued Neural Networks library Barrachina (2021) to work with class functions on SU(3).

We generate 100 discontinuous class functions on SU(3) by taking Gaussian linear combinations of 25 binary valued indicator functions $\mathbf{1}_{k,l}$ for $0 \le k, l \le 4$, where

$$\mathbf{1}_{k,l}(\theta,\phi) = \begin{cases} 1 & \text{if } -3\pi + 1.2k\pi \le \theta \le -3\pi + 1.2k\pi + 1.32\pi \text{ and} \\ & -2\pi + 0.8l\pi \le \phi \le -2\pi + 0.8l\pi + 0.88\pi \\ 0 & \text{otherwise.} \end{cases}$$

Our training data comprises 8500 function evaluations $(\theta, \phi, f(\theta,\phi))$, where $(\theta,\phi)$ was drawn uniformly at random from $[-3\pi, 3\pi] \times [-2\pi, 2\pi]$. To approximate this, we train a fully connected neural network with 6 hidden layers of width 180, 150, 100, 60, 35 and 10 respectively. The input layer consists of the 25 irreducible characters $\chi^{p,q}$ for $0 \le p, q \le 4$. The ReLU non-linear activation function is applied throughout, except in the last layer where the activation function takes the absolute values of complex numbers to provide a real-valued output for the purposes of training. We minimize the squared error loss (MSE) with L2 regularization parameter $\alpha = 10^{-8}$ using AdaMax Kingma & Ba (2017).

As a benchmark, we train a linear regression model with 49 input alphabets $\chi^{p,q}$ for $0 \le p, q \le 7$ and an activation function that takes the absolute values of complex numbers to provide a real-valued output for the purposes of training. We minimize the squared error loss with L2 regularization parameter $\alpha = 10^{-9}$ using AdaMax Kingma & Ba (2017).

Note that we apply the same hyperparameters (chosen to ensure reasonable convergence) for each of the 100 class functions that we train. Our neural network was trained over 100 epochs using a batch size of 500 for each class function. The linear regression model was trained over 300 epochs using a batch size of 250. Our models are evaluated on 1500 unseen datapoints $(\theta, \phi)$ drawn uniformly at random from $[-3\pi, 3\pi] \times [-2\pi, 2\pi]$.

Recording the squared error loss (MSE) of our two models on these 100 class functions, we obtain 200 entries for the histogram in Figure 4. Our neural network performs 160 times better than linear regression with almost double the size of the alphabet, achieving an MSE of just 2.362 as compared to 378.1.

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
