# OpenReview forum: "Conjugation Invariant Learning with Neural Networks"
_ICLR.cc/2022/Conference — ICLR 2022 Submitted_

### Official Review · Reviewer_qL2S · 2021-10-23

**Correctness:** 3
**Technical Novelty And Significance:** 2
**Empirical Novelty And Significance:** 2
**Recommendation:** 3
**Confidence:** 1

**Main Review:**

The presentation of the paper could be significantly improved. In fact, there is no "Introduction" section that describes the context of the problem setting and introduces the contributions of this work at a high level. I don't think it's a good idea to start a paper directly with almost two pages of related works without first introducing the context properly.

Overall, I find the paper hard to follow. Although I tried to understand the contributions of this paper at my best effort, I couldn't properly evaluate its contributions. I understand this may be due to my limited knowledge of group theory, but I also think the authors should try to present the paper in a way that a broad audience in the machine learning community could understand and appreciate. Hence, I would vote for rejection for now, but I encourage the authors to resubmit to the next venue after improving the presentation of the work.


**Summary Of The Paper:**

I have to confess that I was not able to understand the main thesis or contributions of this paper.


**Summary Of The Review:**

I didn't follow the main thesis that this paper tries to convey, and I guess this may happen to a broad audience of the general ML community as well. In my opinion, the main goal of the publication is to spread the idea so that more people in the community could appreciate it. From this perspective, I have to vote for rejection for now.

---

### Official Review · Reviewer_zG1H · 2021-11-04

**Correctness:** 3
**Technical Novelty And Significance:** 2
**Empirical Novelty And Significance:** 1
**Recommendation:** 3
**Confidence:** 4

**Main Review:**

Strengths: By using the theory of characters and class functions, they have created a new method for learning class functions.

Weaknesses:
-The fact that the class of characters must be fixed in order to construct the algorithm.
-The point that when the group is not a finite group, the set of characters is not finite and therefore not sufficiently expressive.
-What seems to be a lack of applications for key machine learning tasks such as graphs and point clouds.

This method is a so-called feature extraction type algorithm. The main problem is that when you try to implement this algorithm by yourself, you have to calculate the characters by yourself, and the set of characters is not finite in general. In recent years, it has become desirable for deep learning to automatically learn the character part of this method and construct a model. Isn't it possible to define a layer that controls the characters and automatically learn the necessary characters for the desired function?

**Summary Of The Paper:**

This paper provides an algorithm for learning class functions that are invariant to special group actions.

**Summary Of The Review:**

By using the theory of characters and class functions, they have created a new method for learning class functions.
However, the scope of application is limited and its usefulness is somewhat questionable, so I judged it not worthy of high evaluation.

---

### Official Review · Reviewer_Y8tG · 2021-11-05

**Correctness:** 4
**Technical Novelty And Significance:** 2
**Empirical Novelty And Significance:** 1
**Recommendation:** 3
**Confidence:** 4

**Main Review:**

The starting point for this paper is recent work on constructing models that respect some task-specific symmetries, as encapsulated by equivariance and invariance constraints, most often to the action of a (locally compact) group. The focal point for the paper, however, is an important class of observables captured by class functions: functions on a group that are constant over conjugacy classes, and thus by extension the paper has relevance to problems that involve learning conjugation invariant functions. Such functions can also be lifted easily to spaces over which the group acts transitively -- a situation more common in problems of interest in machine learning. The conjugation action can be seen as affecting a change of basis and has close connections to a special case of non-commutative Fourier analysis, which is realized by an orthogonal basis of irreducible characters.

One setting in which invariance to such actions arises naturally in ML is covered in an averaging paradigm, encapsulated in proposition 2.1. This is illustrated by various examples quite crisply. Some examples include image alignment in the challenging case when two images lie on a sphere; the QAP; multi-object tracking; an important statistic for comparative analysis of rankings. In particular, it is shown that these problems rely crucially on conjugation invariance and thus on modeling the class function of interest. The paper then makes the claim that it is important to approximate such class functions efficiently since they are clearly relevant to many problems of interest and focuses on it in the sequel.

The paper capitalizes on the notion that irreducible characters are closed under polynomial combinations. This has the implication that using polynomials of a few irreducible characters (of lower order) one can express characters of considerably higher order. The canonical polynomial algebra structure on the characters is handed over by the theory of tensor products of group representations and involves using the Clebsch-Gordan coefficients. As might be apparent, this hints at a general paradigm for approximating general class functions via non-linear, polynomial combinations using a small alphabet of characters. The algorithm is stated in section 4.3. It consists of considering a small number of irreducible characters of the group as input and a set of samples with the group elements and the class function of interest as the label. If the number of irreducible characters is w, then for each example, we construct a w dimensional vector of characters. This w dimensional vector becomes an input to the neural network. For any new input, we first find the w dimensional vector, do forward propagation, and output the result. It is also shown that for groups such as SO(3) and SU(3) a very small number of characters are sufficient to generate all the characters, leading to a good approximation of general class functions. This is illustrated in sections 5.1 and 5.2. Numerical experiments are shown to corroborate the claim.

In my view, the paper presents an interesting direction of research. I found the initial parts of it exciting and enlightening. It indeed hits on an important structure in various machine learning problems that are rarely attacked from the point of view of approximating conjugation invariant functions (although they do appear implicitly in many forms). It must be noted that some problems do use some of the ideas in an implicit form. For instance, the early spectral graph convolution methods can be seen as a special case of convolution with a class function (thus every Fourier component is a sum of characters), the rest of the network can be seen as taking combinations of these outputs. Work on using invariant polynomials (such as by Ward Haddadin are also relevant). However, these works don't directly deal with the scenario that the paper deals with. This paper can also be seen as extracting invariant features, which should be useful for some task of interest.

However, while interesting, the paper needs to pick one of the problems that are stated as examples, and show that the method used here leads to good performance experimentally -- I have a feeling that it might be difficult to get things to work on a problem like solving the QAP. While I liked the paper overall (especially its general discussion in the first half), I think it is not quite ready for publication without extensive experimental results. Only showing that the class function is approximated well by a non-linear network is great, but is not surprising due to the underlying theory. The paper can be made quite strong, and become a general and useful reference if this procedure is shown to be crucial for solving one of the problems listed.

**Summary Of The Paper:**

The paper first makes the case that learning conjugation invariant functions is of interest in many machine learning problems, stating in a general form in which they appear. It then presents a simple method, using a MLP, for approximating such functions efficiently using a small number of irreducible characters. This capitalizes on the notion that irreducible characters are closed under polynomial combinations. Thus, using polynomials of a few irreducible characters (of lower order) one can express characters of considerably higher order. It is shown that the method proposed can approximate class functions of interest well numerically.

**Summary Of The Review:**

The paper presents an interesting perspective of learning conjugate invariant functions. It provides a simple method for the non-linear approximation of class functions. The attendant discussion for many problems of interest is also valuable and useful. The paper does not propose a new theory -- it crisply teases out the importance of building in such invariance to many problems of interest. While valuable, it needs validation on one problem of interest. I think the paper has the potential to be a strong and valuable contribution.

---

### Official Review · Reviewer_E1Av · 2021-11-07

**Correctness:** 4
**Technical Novelty And Significance:** 4
**Empirical Novelty And Significance:** 1
**Recommendation:** 5
**Confidence:** 2

**Main Review:**

The paradigm proposed in the paper is very interesting, and I liked the manuscript, especially the Appendix, which explains the mathematical concepts in an easy-to-understand manner.
Personally, I feel the paper very interesting and instructive, and I think it should definitely be published somewhere.
On the other hand, as the authors mentioned in 7 Discussion, the paper does not mention the application of the proposed paradigm around machine learning field.
Also, it was not clear how the proposed paradigm would solve the problem of Image Alignment, which was mentioned in the paper as a possible application.
These facts may indicate that this paper is a little different from the scope of this conference.
Therefore, it seems inappropriate for this paper to be accepted for the conference.

In addition, the main text of the paper, especially the introduction part, was difficult for me to understand how each paragraph relates to the preceding and following paragraphs.
Also, the description using only sentences hindered my understanding.
It would be desirable to use equations and figures to make the description more understandable to people from various fields in this conference.

**Summary Of The Paper:**

In this manuscript, the authors develop a DNN paradigm that learns functions on groups as polynomials using canonical tracial class functions as an orthogonal basis for class functions called irreducible representations.
Since the basis constitutes a nonlinear coordinate system, it has the potential to efficiently handle complex symmetries.
The paper shows analyticaly and numericaly that for simple systems such as SO(3) and SU(3), the proposed paradigm has high expressive power with few bases.

**Summary Of The Review:**

As the authors mentioned, the paper does not mention the application of the proposed paradigm around machine learning field.
Also, it was not clear how the proposed paradigm would solve the problem of Image Alignment, which was mentioned in the paper as a possible application.
These facts may indicate that this paper is a little different from the scope of this conference.

---

### Official Review · Reviewer_5x3M · 2021-11-07

**Correctness:** 4
**Technical Novelty And Significance:** 3
**Empirical Novelty And Significance:** 2
**Recommendation:** 3
**Confidence:** 4

**Main Review:**

### Strengths
The paper addresses learning with invariances which is an important topic, and a white-hot one. Looking at class functions is natural and the paper does a good job of reviewing the various situations in computational sciences where class functions arise. The authors convincingly argue that higher-order characters can be obtained as functions of low-order characters and that as a consequence neural networks may be able to effectively learn complicated class functions. The paper is well written and easy to follow, it does a good job summarizing the requisite group theory, and the numerical experiments indicate the potential of the proposed strategy.

### Weaknesses

- The strategy proposed by the authors follows a standard recipe borne out in numerous machine learning and signal processing contexts: compute a set of _fixed_ invariant features (with a suitable notion of invariance) before applying a standard learning model (here a neural network). The only technical content in the manuscript is on pages 7 and 8. There is absolutely nothing wrong with simplicity, brevity, and strong introductions (to the contrary), but in this case I cannot escape the impression that the idea is good but the paper needs a lot of work (either on the side of theory, or the numerics, or both). In particular, the motivations from the introduction were not at all vindicated in the experiments.

- The two theorems on page 7 state that the polynomial algebras of characters of SO(3) and SU(3) are generated by a few irreducible characters. This fact is not surprising to people familiar with group representations (I would like to say that it is known although (perhaps) not commonly written in this exact form. It certainly follows directly from (3) in the case of SO(3) and (5) for SU(3).) It is still a good motivation for applying neural nets to a small set of features obtained from low-order irreducible characters, but one would hope that this discussion will be taken much further. For instance, for common class function, when can we expect good approximation properties of this scheme? This would require looking into how the used neural networks generate high frequencies from low frequencies (in the context of irreducible characters) or some other strategy to establish strong approximation statements.

- The introduction describes several exciting motivations but I find these descriptions somewhat disconnected from what could and should be learned. The numerical experiments on page 8 only involve direct synthetic constructions of class functions and tackle none of these motivating applications.

- In the motivation via image alignment, where do you see your scheme playing a role? The quantity $\mathcal{M}_{x, y}$ is indeed a distance and not a class function itself; $\tilde{f}$ _is_ a class function but it can typically be computed exactly and very fast using FFTs on groups or homogeneous spaces. Can you provide a related numerical example, even if stylized, where your strategy can improve things?

- In the motivation via QAP, you suggest that learning $f$ gives a stopping algorithm. Can one in general hope to learn $f$? What is its complexity class? Also, for any reasonable $\epsilon$, why would it be possible to do well using gradient descent on this very non-convex problem?

- Figures 1 and 2 have no caption which makes it hard to interpret them.



**Summary Of The Paper:**

This paper proposes a strategy to learn class function on groups (and by extension on homogeneous spaces). The paper is motivated by applications in inverse problems and image processing (e.g. particle alignment in CryoEM) as well as integer programming (e.g. the quadratic assignment problem). The authors note that irreducible characters form a basis for square-integrable class functions and then apply a neural network on a set of invariant features chosen as low-order irreducible characters rather than directly on the input, which naturally results in a conjugation-invariant function. They argue that the neural network (thanks to the nonlinear activations functions) can generate the needed high-order characters. The obtained invariant architecture is then applied to several toy problems with synthetic data.


**Summary Of The Review:**

This paper proposes a nice idea: to learn class functions by first applying a feature transform, where features are chosen to be invariant to conjugation (in particular, low-order irreducible characters), and then processing those by a neural network. The authors justify this by noting that irreducible characters form an orthobasis for the space of square-integrable class functions, and that (by their algebraic structure), high-order characters can be generated from a few low-order characters. The paper is well written but at this conference I would expect to see much more technical content, stronger results, more convincing numerical experiments. I think that this can become a strong paper but the current version records a nice idea with solid potential and presents some initial numerics.

---

### Decision · Program_Chairs · 2022-01-20

**Decision:**

Reject

**Comment:**

The reviews received for this paper raise several critical concerns to which the authors have not provided a response. Thus, in its present form, the paper is not ready for publication.